# An Enhanced Saline Soil Dielectric Constant Model Used for Remote Sensing Soil Moisture and Salinity Retrieval

Liang Gao [1,2], Xiaoning Song [1,2,*], Xiaotao Li [3], Jianwei Ma [3], Pei Leng [4,5], Weizhen Wang [6] and Xinming Zhu [1,2]

1   College of Resources and Environment, University of Chinese Academy of Sciences, Beijing 101408, China; gaoliang17@mails.ucas.ac.cn (L.G.); zhuxinming19@mails.ucas.ac.cn (X.Z.)
2   Yanshan Earth Critical Zone and Surface Fluxes Research Station, University of Chinese Academy of Sciences, Beijing 101408, China
3   China Institute of Water Resources and Hydropower Research, Beijing 100038, China; majw@iwhr.com (J.M.)
4   Key Laboratory of Agricultural Remote Sensing, Ministry of Agriculture and Rural Affairs, Beijing 100081, China; lengpei@caas.cn
5   Institute of Agricultural Resources and Regional Planning, Chinese Academy of Agricultural Sciences, Beijing 100081, China
6   Northwest Institute of Eco-Environment and Resources, Lanzhou 730000, China; weizhen@lzb.ac.cn
*   Correspondence: songxn@ucas.ac.cn

**Abstract:** The soil dielectric constant model is essential for retrieving soil properties based on microwave remote sensing. However, the existing saline soil dielectric constant models perform poorly in simulating the dielectric constant of soil with high water content and salinity. In this study, the Wang Yueru (WYR) saline soil dielectric constant model, which was demonstrated to perform well in describing the effect of salinity and moisture on the dielectric constant, was validated based on experimental measurements of soil samples under different water content and salinity degrees. Furthermore, we adjusted the model form, refitted the empirical coefficient in the model, and finally acquired a two-stage model for simulating the soil dielectric constant. The enhanced model was validated under different soil moisture and salinity ranges using experimental measurements of soil samples. Compared to the original model, the proposed model exhibits a larger improvement in simulating the soil dielectric constant, and the RMSE of the simulated results dramatically decreased from 7.3 to 1.6, especially for soil with high salinity and water content. On this basis, a model suitable for L-band microwave was established. This model is of great significance for studying soil dielectric characteristics and retrieving soil parameters based on L-band data. Furthermore, this model can be used to retrieve soil salinity and water content using microwave remote sensing under a broadened application situation, such as in saline-alkali soils, wetlands, and salt marshes.

**Keywords:** saline soil; soil moisture; salinity; dielectric constant; high salinity and water content

## 1. Introduction

Soil moisture (SM) and salinity are essential parameters in surface ecosystems [1–3]. These two soil components provide the basic materials for the physiological activities of creatures and vegetation [4,5]. Soil moisture takes part in the surface circulation of surface energy and matter. As a key part of the Earth's surface water system, soil moisture is a source of surface evaporation and groundwater infiltration. At the same time, soil moisture is critical as a sink of precipitation and groundwater recharge [6,7]. The amount of soil moisture influences the exchanges of energy and matter between the surface environment and the soil [8–10]. Moreover, salinity is an essential soil parameter influencing land surface processes, such as the freezing and thawing cycle [11,12] and cultivated land salinization [3,13]. In addition, irrigation is one of the main factors to consider for improving production [14–16]. Rapid increases in the spatial distribution and frequency of irrigation have contributed to the rapid increase in soil salinization in recent decades [13]. Soil

salinization leads to the loss of soil fertility and is one of the primary threats to global agricultural production and food safety [17,18]. Therefore, monitoring soil moisture and salinity is essential for eco-environmental scientific research and agricultural production.

Remote sensing provides an effective approach to obtaining spatial soil moisture and salinity information based on different retrieval algorithms [19–21]. Among various remote sensing data sources, microwaves have vast advantages in observing the Earth's surface [21,22]. On the one hand, microwaves have a longer wavelength than visible light and can penetrate clouds and rainy weather. This makes microwaves an efficient way of observing the Earth's surface [23]. Secondly, microwaves are sensitive to changes in soil moisture and salinity [24,25]. Soil properties such as water content, salinity, and soil texture commonly determine the active microwave backscatter coefficient by influencing the soil dielectric constant [26,27]. The soil dielectric constant model is an important part of the active microwave soil moisture and salinity retrieval model.

Many empirical and theoretical models that can simulate the complex dielectric constant of soil with several soil properties have already been developed [28–30]. Generally, empirical models have been established based on many experimental measurements by fitting the relationships between soil permittivity and soil properties [28,31]. These soil properties usually include the volumetric water content, soil texture, bulk density, specific density, and temperature, etc. Physical models are generally built based on the refractive volumetric mixing model, and the complex dielectric constant of mixed soil is represented as the weighted sum of each component [31,32]. Because the model expression is complex and the parameters are challenging to obtain, most physical models are difficult to realize in some specific applications. Therefore, many semi-empirical models have been built based on physical models and experimental measurements [28,31,33,34]. Among many semi-empirical models, the Wang [31], Mironov [32], and Dobson [28] models are widely used to simulate dielectric constant and retrieve soil properties based on microwave remote sensing.

Many studies have demonstrated that salinity greatly influences soil dielectric characteristics [24,25]. However, the aforementioned models rarely consider the impact of soil salinity on the soil dielectric constant [3,35]. As typical semi-empirical models, the Dobson salinity (Dobson-s) model, Wang Yueru (WYR) model, and Hu Qinrong (HQR) model have been widely used to simulate the dielectric constant of saline soil [28,36,37]. The Dobson-s model represents the free water dielectric constant in the original Dobson model using the saltwater dielectric constant. In the original Dobson model, the water dielectric constant is calculated by the Debye model. Parameters such as the high-frequency limit of the dielectric constant, effective conductivity, and static dielectric constant are the main factors used to determine the dielectric constant of water [24,38,39]. Another famous model is the Stogryn equation [39], in which salinity is related to an empirical coefficient. In this model, the normality salinity is used as an intermediate variable to qualitatively describe the soil salinity information, and the empirical coefficient is mainly related to the normality salinity. Many studies have demonstrated that the Stogryn model has an error in simulating the saltwater dielectric constant. Klein and Swift [38] updated the saltwater dielectric constant based on the Stogryn model. They built an empirical equation directly between the salinity and dielectric constant of saltwater. Many studies have concluded that the Dobson-s model underestimates the imaginary part of the saline soil dielectric constant, and the primary source of uncertainty comes from the imaginary part of the saltwater dielectric constant [36,37]. The conductivity of the saline solution is the critical parameter influencing the imaginary part of the saltwater dielectric constant [35], and the latter model has rewritten this parameter. Hu et al. [36] modified the expression of conductivity in the Debye model based on Stern–Gouy theory and considered the surface area of solid soil particles. Later research demonstrated that the model has a poor performance when the volumetric water content is less than 0.3 $m^3/m^3$ [35,37]. Wang et al. [37] concluded that there is a linear relationship between the conductivity of the saline solution and the concentration of ions. The model assumes that the concentration of ions of the soil's saline

soil solution increases with the salinity content for a specific water content, and the concentration of ions in saline soil solution is generally inversely proportional to the soil water content. Based on the aforementioned two phenomena, the WYR model builds the empirical relationship between the conductivity of soil solution and soil volumetric, salinity. The empirical coefficients of the model are fitted based on many experimental measurements. Due to the limitations of the model assumptions, these models are applicable under specific conditions. The simulation results have more significant uncertainty when the soil has high a volumetric water content [35]. Specifically, validations using measured data have shown that the error of the model simulation results show an increasing trend with increasing soil moisture and salinity [40]. Thus, it is necessary to improve the model to allow broadened application situations, especially for soils with high water contents, based on the existing models and experimental measurements.

The main objective of our study is to establish a saline soil dielectric constant model that can improve the simulation ability of the relationships between the saline soil dielectric constant and the soil moisture, salinity, and soil texture. Specifically, the existing saline soil dielectric constant models are validated using experimental measurements, and the performance of the models is analyzed during the different application conditions. Based on the soil dielectric phenomenon from experimental measurements, we aim to develop an enhanced model that has a better performance in simulating the saline soil dielectric constant under broadened salinity and water content conditions.

## 2. Sample Dataset and Models

### 2.1. Soil Sample Dataset

Many soil samples were collected in this study, and the dataset was obtained from [35,37]. The dataset includes five soil samples: soil-14, soil-30, soil-36, soil-37, and soil-5. The properties of these soil samples, including the bulk density, specific density, and soil texture, were measured in the laboratory to create the dataset. These soil samples were collected from the middle reaches of the Heihe River, one of the largest inland rivers in Northwest China. Saline soil samples with various water contents and salinities were organized in the laboratory. The soil moisture and salinity of the samples contain many degrees. Specifically, the NaCl solutions with different concentrations were put into dry soil to form soil samples with different salinity contents, and the different soil samples had salinity degrees of 3, 6, 9, 12, 15, 18, 20, 40, and 100 g/kg. At the same time, the soil samples were configured with volumetric water contents of 5%, 10%, 20%, 30%, and 40%. Each soil sample contained a combination of all the soil moisture and salinity content profiles described above, and each of these combinations was measured three times. Finally, the complex dielectric constant of the saline soil samples was measured using a vector network analyzer (VNA), and the specific measurement equipment and sample preparation can be found in [35,37]. The VNA used the dielectric probe to measure the complex dielectric constant of mixed soil, and this instrument has an electromagnetic wave signal frequency ranging from 5 MHz to 20 GHz. The real and imaginary parts of the complex dielectric constant for each soil sample were measured at several frequencies (from 0.2 to 20 GHz). During the data preprocessing, we ignored the abnormal measurements based on the three-sigma rule [41,42] and regarded the average value of several measurements as the true value of the saline soil dielectric constant. The specific approach is as follows:

$$\varepsilon'_{soil} = \frac{\sum_{i=1}^{n} \varepsilon'_{soil,i}}{n} \tag{1}$$

$$\varepsilon''_{soil} = \frac{\sum_{i=1}^{n} \varepsilon''_{soil,i}}{n} \tag{2}$$

where the $\varepsilon'_{soil}$ and $\varepsilon''_{soil}$ are the real and imaginary parts of the mixed soil dielectric constant, respectively, $\varepsilon'_{soil,i}$ and $\varepsilon''_{soil,i}$ are the $i$-th observed real and imaginary parts of the mixed soil dielectric constant, and $n$ is the observation number after removing abnormal measurements.

### 2.2. Saline-Soil Dielectric Constant Model

Many physical and semi-empirical models have already been built to simulate the impact mechanisms of salinity and other soil properties on the saline-soil dielectric constant [28,35,36,43]. The Dobson model first considers the salinity content in the mixing soil dielectric constant model and has an excellent performance in simulating the salinity impact [28]. The original Dobson model is a semi-empirical model that was proposed based on experimental measurements and has an application range from 1.4 to 18 GHz. The model was established on the physical basis of the refractive volumetric mixing model. It assumes that the total complex dielectric constant of mixing soil is the sum of the total composition, including soil particles, bound water, free water, and air.

The specific expression of the Dobson model is as follows:

$$\varepsilon_{soil}^{\alpha} = 1 + \frac{\rho_b}{\rho_s}\left(\varepsilon_s{}^{\alpha} - 1\right) + m_v{}^{\beta}\varepsilon_{sw}{}^{\alpha} - m_v \tag{3}$$

where $\varepsilon_{soil}$ is the soil dielectric constant, $\rho_b$ is the soil bulk density, $\rho_s$ is the specific density, $\alpha$ is an empirical coefficient related to soil type and is usually considered to be 0.65, $m_v$ is the volumetric water content of the soil, $\varepsilon_s$ is the dielectric constant of solid soil particles, and $\varepsilon_{sw}$ is the saltwater dielectric constant. For the aforementioned theoretical model, the real part ($\varepsilon'_{soil}$) and the imaginary part ($\varepsilon''_{soil}$) of the saline-soil complex dielectric constant can be respectively expressed as follows:

$$\varepsilon'_{soil} = \left[1 + \frac{\rho_b}{\rho_s}\left(\varepsilon_s^{\alpha} - 1\right) + m_v^{\beta'}\varepsilon_{sw}'^{\alpha} - m_v\right]^{1/\alpha} \tag{4}$$

$$\varepsilon''_{soil} = \left[m_v^{\beta''}\varepsilon_{sw}''^{\alpha}\right]^{1/\alpha} \tag{5}$$

where $\varepsilon'_{soil}$ and $\varepsilon''_{soil}$ are the real and imaginary parts of the saline-soil complex dielectric constant, respectively, $\beta'$ and $\beta''$ are the empirically determined parameters related to soil texture, $\varepsilon'_{sw}$ and $\varepsilon''_{sw}$ are the real and imaginary parts of the saltwater dielectric constant, respectively. In the original Dobson saline soil model, the expression of the real and imaginary parts of the free water dielectric constant is given by the Debye model. For the Dobson saline soil model, these two parameters are represented by the dielectric constant of saltwater, and the specific expression is as follows:

$$\varepsilon'_{sw} = \varepsilon_{sw\infty} + \frac{\varepsilon_{sw0} - \varepsilon_{sw\infty}}{1 + \left(2\pi f \tau_{sw}\right)^2} \tag{6}$$

$$\varepsilon''_{sw} = \frac{2\pi f \tau_{sw}\left(\varepsilon_{sw0} - \varepsilon_{sw\infty}\right)}{1 + \left(2\pi f \tau_{sw}\right)^2} + \frac{\sigma_{sw}}{2\pi f \varepsilon_0} \tag{7}$$

where $f$ is the frequency of microwaves and $\varepsilon_0$ is the permittivity of the free space. In the aforementioned expression for the saltwater dielectric constant, there are four parameters: the electrostatic field permittivity of saltwater ($\varepsilon_{sw0}$), the relaxation time of saltwater ($\tau_{sw}$), the high-frequency extreme value of the dielectric constant of saltwater ($\varepsilon_{sw\infty}$), and the conductivity of the soil solution ($\sigma_{sw}$). These parameters determine the dielectric constant of saltwater together. For these saltwater-related parameters, many studies have already provided theoretical and empirical approaches for their calculation [37–39]. Specifically, the high-frequency extreme value of the dielectric constant of saltwater ($\varepsilon_{sw\infty}$) is regarded as having an identical value to the parameters of free water and is considered to have a specific value of 4.9 [44]. The Stogryn model gives semi-empirical formulas to acquire

$\varepsilon_{sw0}$ $\tau_{sw}$ and $\sigma_{sw}$ [39]. The normality of the solution was used as an indirect parameter to simulate the influence of soil salinity, and the quantitative relationship between this variable and soil salinity was provided in this model. Klein has demonstrated many differences between the measurements and the simulation results of $\varepsilon_{sw0}$, $\tau_{sw}$ and $\sigma_{sw}$. They adjusted the Stogryn model and directly established the relationships between these three parameters and salinity with the specific expression provided here [38].

Many studies have demonstrated that the Dobson-s model underestimates the imaginary part of the saline soil complex dielectric constant [35,37,40]. Many adjusted models have been proposed based on many measurements and the original Dobson-s model to build models that have an excellent performance for simulating the impact of salinity on the soil dielectric constant. Among them, the WYR model has great accuracy when simulating the saline soil dielectric constant through validation using experimental measurements [35]. It assumes that the concentration of ions ($S_{mv}$) gradually decreases with increasing soil moisture content ($m_v$), with the following relationship:

$$S_{mv} = b\rho_b \frac{S_c{}^{m_v{}^{\alpha}}}{m_v} \tag{8}$$

where $Sc$ is the soil salinity content, the b and $\alpha$ are the empirical coefficients used to represent this relationship. In addition, some studies have demonstrated that there is a linear relationship between the concentration of ions of the soil solution ($S_{mv}$) and the conductivity ($\sigma_{sw}$), as follows:

$$\sigma_{sw} = aS_{mv} \tag{9}$$

where $a$ is an empirical coefficient. Hence, the conductivity ($\sigma_{sw}$) can be represented as follows:

$$\sigma_{sw} = ab\rho_b \frac{S_c{}^{m_v{}^{\alpha}}}{m_v} = c\rho_b \frac{S_c{}^{m_v{}^{\alpha}}}{m_v} \tag{10}$$

where $a$ and $b$ are rewritten as empirical coefficient $c$. Finally, the conductivity ($\sigma_{sw}$) is expressed by the aforementioned assumption in the original WYR model, and the specific expression of the imaginary part of the saltwater dielectric constant is as follows:

$$\varepsilon_{sw}'' = \frac{2\pi f\tau_{sw}(\varepsilon_{sw0} - \varepsilon_{sw\infty})}{1 + (2\pi f\tau_{sw})^2} + c\frac{\rho_b}{2\pi f\varepsilon_0}\frac{S_c{}^{m_v{}^{\alpha}}}{m_v} \tag{11}$$

Based on measurements obtained under the C-band (5.3 GHz), the model acquired the specific expression, and empirical coefficients $c$ and $\alpha$ have values of 0.371 and 0.18, respectively [37].

Many studies have already demonstrated that salinity significantly influences the imaginary part of the saltwater dielectric constant [44,45]. However, the existing models have large errors when simulating this variable, especially for soil with high water and salinity contents [37]. Therefore, we focus on the imaginary part model component corresponding to the saltwater dielectric constant in this study. For the WYR model, some detailed validations have been implemented to test its performance under different salinity and moisture situations at the C-band (5.3 GHz). The root mean square error (RMSE) and coefficient of determination ($R^2$) are used to indicate the model performance. The measured value is the experimentally measured imaginary part of the dielectric constant for soil with specific moisture and salinity; this value is considered the true value of the saline soil dielectric constant. The predicted value is the imaginary part of the dielectric constant simulated by the WYR model.

Figure 1 compares experimental measurements and model-simulated saline soil dielectric constants under different soil salinity and moisture degrees. When the soil salinity is more than 20 g/kg, Figure 1 shows that a significant difference exists between the original WYR model-simulated imaginary part and the experimental measurements (pink and blue points in Figure 1b). Through the evaluation based on the experimental measure-

ments, we concluded that the WYR model performs poorly when the soil salinity is greater than 20 g/kg. This phenomenon occurs across all soil moisture ranges, specifically from 0.05 to 0.4 m$^3$/m$^3$. In addition, the response of the model to soil salinity is related to the soil water content. When the soil water content is greater than 0.2 m$^3$/m$^3$, the model has an enormous uncertainty when simulating soils with salinity reaching 20 g/kg. However, the model still performs well when the soil water content is 0.1 m$^3$/m$^3$, and the salinity is greater than 40 g/kg. In addition, greater differences occur between the simulated and measured values when soil moisture remains fixed, and salinity is relatively high, as is the case for the green points in Figure 1a ($Mv$ = 0.05 m$^3$/m$^3$). Thus, it could be found that these points deviate from the 1:1 line when salinity increases in Figure 2. This phenomenon is also suited to soil salinity. The evaluation results demonstrate that the model is unsuitable for soil with high salinity, and the effect of salinity on the model is related to the soil water content.

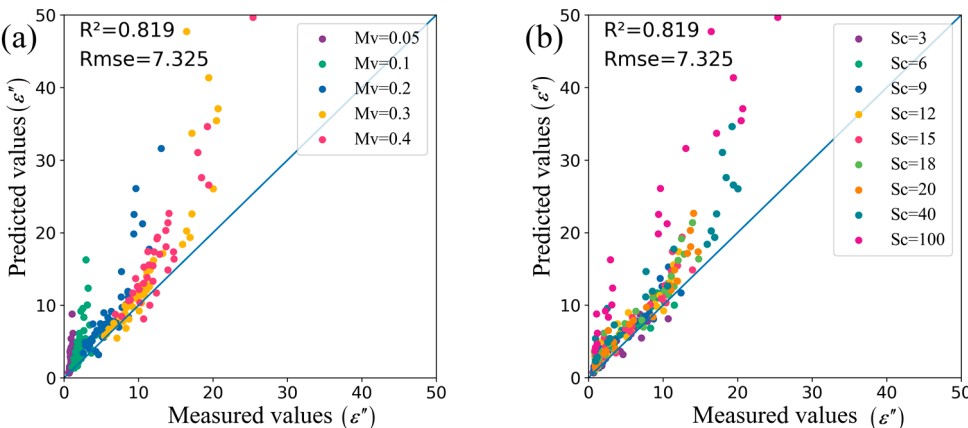

**Figure 1.** The model-simulated result validation using soil samples under different (**a**) soil moisture ($Mv$, m$^3$/m$^3$) and (**b**) salinity ($Sc$, g/kg) classifications.

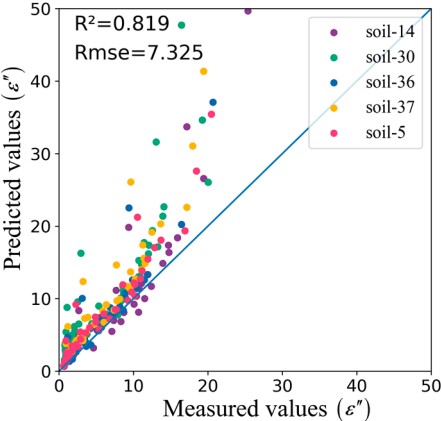

**Figure 2.** The model-simulated result validation under different soil samples.

Soil texture is another property that determines the soil dielectric constant. The soil samples that validate the model have different soil textures and compositions. To test the responses of the saline soil dielectric constant model to soils with different soil textures, the WYR model was validated based on experimental measurements with separate soil samples. Figure 2 shows that the difference between the simulated results and measurement has a similar statistical distribution for different soil samples. Specifically, the simulated dielectric constant has a larger error when the value is greater. The validation from separate soil samples demonstrates that the simulated result for responding soil textures had no significant errors. The main source of the difference between the simulated results and

measurements is that the model did not adequately simulate the relationship between the soil dielectric constant and soil moisture salinity content, especially for the imaginary part.

To further improve the simulation ability of the model for the imaginary part of the soil dielectric constant, this study refined the model based on the existing model assumptions and experimental measurements. Finally, we developed a semi-empirical model that performs well when simulating the impact mechanisms of salinity and soil moisture on permittivity.

### 2.3. Improved Model in C-Band (5.3 GHz)

The validation shows that the original WYR model has a larger error in simulating the imaginary part of the saline soil dielectric constant in situations when the soil moisture exceeds 0.2 m$^3$/m$^3$ and salinity exceeds 20 g/kg. For this characteristic, the assumptions and applicable conditions of the model need to be further analyzed and explored. The original WYR model assumes that the concentration of ions undergoes a slow rate of decrease [37]. The speed of the concentration of ions decreases greatly when the soil has a high salinity, causing the previous assumption to become unsuitable. In addition, the HQR model builds a linear relationship between salinity and soil conductivity [36]. Some studies have demonstrated that the HQR model has an excellent performance in simulating the dielectric constant when saline soil has high water content and salinity [35,37]. Therefore, based on the above two existing models, we modify the expression of the ions concentration, aiming to obtain a model that performs better under broadened application conditions. The enhanced model further represents the effective conductivity using the new relationship in the original model about the imaginary part of saltwater. Because of the different assumptions, the model has different expressions under different soil moisture and salinity ranges. Specifically, the model assumes that there exists a power function between the effective conductivity and salinity when soils have high salinity and water contents, and the exponent is a constant (see Equation (12)). For soils with other salinity and water content conditions, the improved model is consistent with the form of the original WYR model; the aforementioned exponent is related to soil water content (see Equation (13)). Therefore, a two-stage model is established to describe the effects of salinity and soil moisture on the saline soil dielectric constant. The empirical coefficients in the model are fitted based on the measured data in different application conditions, and in this part, the nonlinear least squares approach was used [46]. The saltwater dielectric constant can be expressed as follows:

If the soil moisture is more than 0.3 m$^3$/m$^3$ and salinity is more than 20 g/kg:

$$\varepsilon''_{sw} = \frac{2\pi f \tau_{sw}(\varepsilon_{sw0} - \varepsilon_{sw\infty})}{1 + (2\pi f \tau_{sw})^2} + 1.244 \frac{\rho_b}{2\pi f \varepsilon_0} \frac{S_c^{0.376}}{m_v} \left(m_v \geq 0.3 \text{ m}^3/\text{m}^3 \text{and } S_c \geq 20 \text{ g/kg}\right) \tag{12}$$

Except for the aforementioned range of soil moisture and salinity, the imaginary part of the saltwater dielectric constant can be expressed as follows:

$$\varepsilon''_{sw} = \frac{2\pi f \tau_{sw}(\varepsilon_{sw0} - \varepsilon_{sw\infty})}{1 + (2\pi f \tau_{sw})^2} + 0.378 \frac{\rho_b}{2\pi f \varepsilon_0} \frac{S_c^{m_v^{0.412}}}{m_v} \left(m_v < 0.3 \text{ m}^3/\text{m}^3 \text{ or } S_c < 20 \text{ g/kg}\right) \tag{13}$$

## 3. Results and Discussion

### 3.1. Model Validation

#### 3.1.1. The Accuracy Validation of the Improved Model

The improved model was validated on different soil moisture and salinity degrees (Figure 3). The measured value is the experimentally measured dielectric constant for the soil with specific moisture and salinity, corresponding to the true value of the saline soil dielectric constant. The predicted value is the dielectric constant simulated by the improved model. In Figure 3, the validation results are shown using a scatter diagram in

which different colors display the performance of the improved model under diverse soil salinity and water content degrees.

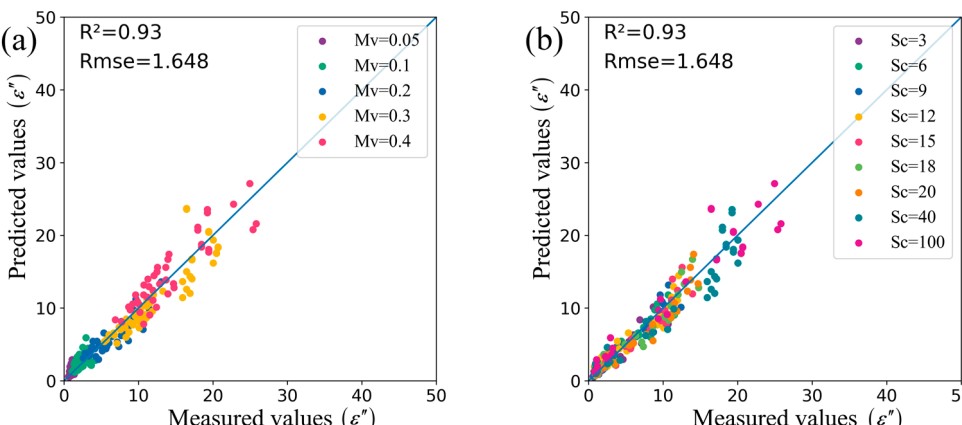

**Figure 3.** The validation of the improved model-simulated results for different (**a**) soil moisture ($Mv$, m$^3$/m$^3$) and (**b**) salinity ($Sc$, g/kg) classifications.

Figure 3 shows that the simulated result of the imaginary part of the saline soil dielectric constant has good accuracy in that the RMSE is less than 1.65, and the R$^2$ is greater than 0.93. Compared to the original WYR model, the RMSE of the improved model dramatically decreased from 7.325 to 1.6. The simulated results are closer to the experimentally measured dielectric constant values under the corresponding conditions. The scatter points between these two parameters are basically distributed around the 1:1 line. In addition, from Figure 1, the validation result shows that the original model performs poorly for soils with high water and salinity contents. Compared to the original model, the improved model has a better ability to simulate the saline soil dielectric constant under high salinity conditions. However, there is still a large difference between the model-simulated results and measurements when the soil has a high salinity and water content. The RMSE of the simulated results in the enhanced model also shows an increasing trend compared to the soil samples with low water and salinity contents.

### 3.1.2. The Accuracy Comparison of the Original Model and Improved Model under Different Soil Moisture Classifications

Except for the aforementioned validation in the global application conditions, a detailed validation was realized under the relatively fine soil moisture ranges (Figure 4). For the original model, with increasing soil moisture, the RMSE of the simulation results increases rapidly. Specifically, the simulation result has an RMSE close to 2.5 when the soil moisture reaches 0.05 m$^3$/m$^3$, and the RMSE is greater than 10 when the soil moisture is 0.4 m$^3$/m$^3$. These results show that the original model has a low accuracy when simulating the dielectric constant of soils with a high soil water content. Compared with the original WYR model, the improved model significantly improved the simulation of the saline soil dielectric constant. For all soil moisture ranges, the enhanced model has a smaller RMSE than the original model. Specifically, when the soil moisture reaches 0.4 m$^3$/m$^3$, the simulated dielectric constant of the improved model has the largest RMSE, at approximately 2.5. This value is close to the minimal RMSE of the original model. In addition, the RMSE of the simulation results exhibits a rising trend as the soil water content increases. The RMSE of the model simulation results is small when the soil moisture is low, and when the soil moisture is high, the RMSE has the opposite performance. This means that the uncertainty of the model simulation results becomes larger as the soil moisture content increases. However, compared with the original model, the model accuracy greatly improved under the same soil moisture content level.

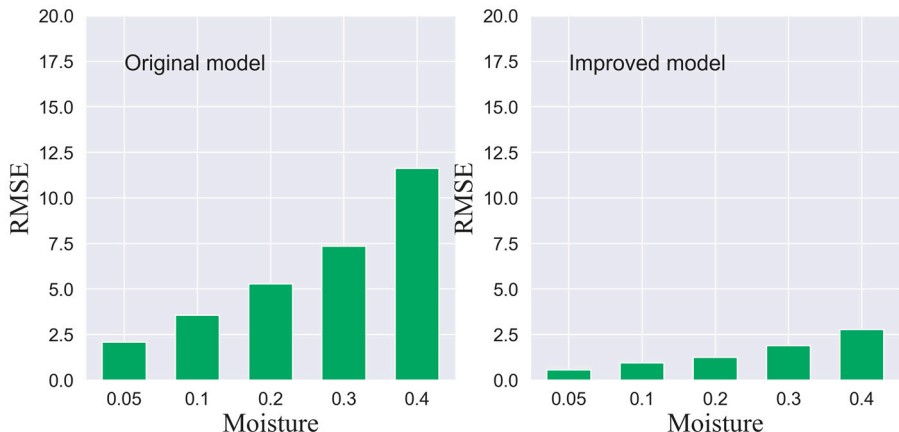

**Figure 4.** The accuracy comparison between the original WYR model and the improved model for different water content (m$^3$/m$^3$) classifications.

### 3.1.3. The Accuracy Comparison of the Original Model and Improved Model under Different Salinity Classifications

Many validations are realized to investigate the performance of the improved model under different salinity conditions in this section. The RMSEs of the original model and the improved model are shown in Figure 5. The result indicates that the original model performs terribly when the soil has a high salinity content. Specifically, the RMSE is greater than 5 when the salinity is greater than 40 g/kg. Under each salinity degree, compared with the original model, the improved model has a good performance, and the RMSEs of the simulated result are less than 2.5 for all salinity ranges. In addition, the simulation capability of the improved model is greatly advanced under high salinity conditions. Specifically, the RMSEs of the simulated result largely decrease when the salinity is greater than 9 g/kg. At the same time, the RMSE of the improved model simulation result has a rising trend as the soil salinity increases. This phenomenon is a reasonable result. When other soil properties are the same, the value of the soil dielectric constant increases with the soil moisture increasing, and low-moisture soil has a smaller dielectric constant, while high-moisture soil has a larger dielectric constant. Generally, the improved model has the same level of relative error under different soil salinity conditions.

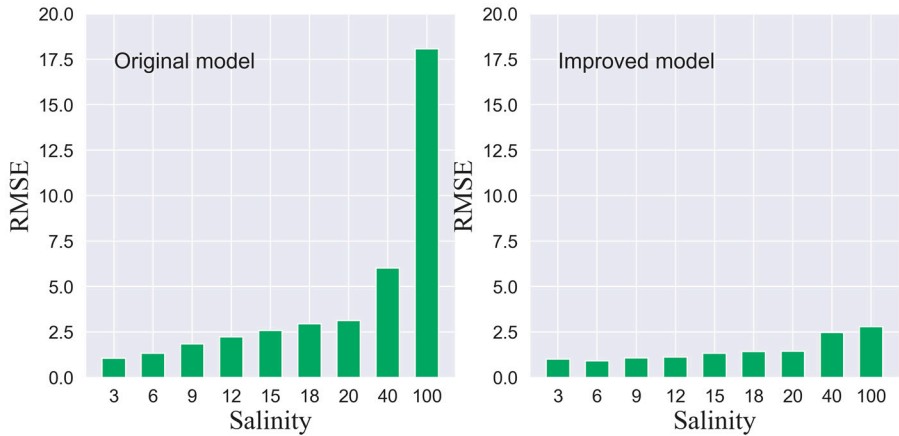

**Figure 5.** The accuracy comparison between the original WYR model and the improved model for different soil salinity (g/kg) classifications.

### 3.1.4. The Accuracy Comparison of the Original Model and Improved Model for Each Soil Sample

The different soil samples have diverse soil textures. The soil texture is also a vital soil property for determining the soil dielectric constant. The accuracy of the simulated

dielectric constant was validated based on individual soil samples to test the model's response to soil textures (Figure 6). Compared with the original model, the validation shows that the improved model has a good performance when simulating the dielectric constant of soils with different soil textures, and the RMSE differences among the diverse samples decrease in the improved model. Specifically, the RMSEs of the original model are greater than 2.5 for each soil sample, and the RMSEs of the improved model are less than this value for all soil samples. In addition, the RMSE of the original model varies greatly among different samples, and the improved model has a similar RMSE for each soil sample. The RMSE of the simulated result has greatly decreased. At the same time, for each soil sample. The improved model generally has an excellent ability to simulate the impact of the soil texture on the saline soil dielectric constant.

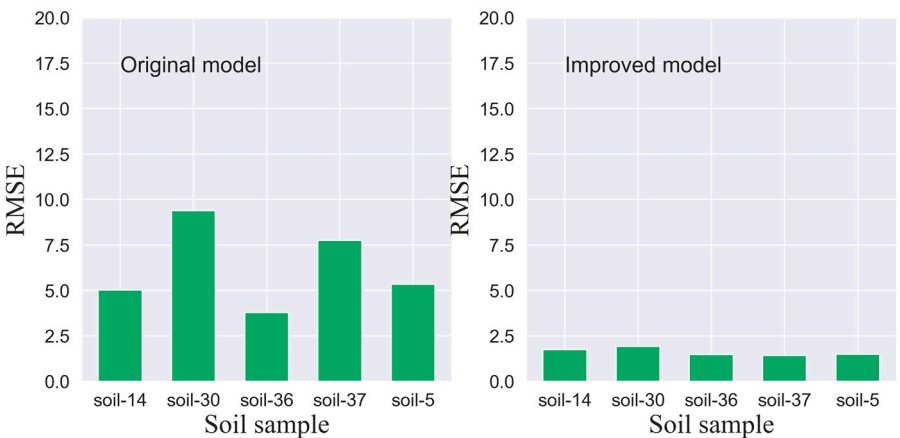

**Figure 6.** The accuracy comparison between the original WYR model and the improved model for different soil samples.

### 3.2. The Model Description at L (1.2 GHz) Band

To expand the applicable range of the model, we further fitted the empirical coefficients and obtained a model that is suitable for the L-band. This model has the same expression as the C-band model, and only the empirical coefficients in the two models are different. The specific expression of the model that is suitable for L-band microwaves is expressed as follows:

When the soil moisture is more than 0.3 m$^3$/m$^3$ and salinity is more than 20 g/kg.

$$\varepsilon''_{sw} = \frac{2\pi f \tau_{sw}(\varepsilon_{sw0} - \varepsilon_{sw\infty})}{1 + (2\pi f \tau_{sw})^2} + 1.765 \frac{\rho_b}{2\pi f \varepsilon_0} \frac{S_c^{0.284}}{m_v} \ (m_v \geq 0.3 \text{ m}^3/\text{m}^3 \text{and } S_c \geq 20 \text{ g/kg}) \tag{14}$$

Except for the aforementioned soil moisture and salinity range, the expression of the imaginary part of the saltwater dielectric constant is as follows:

$$\varepsilon''_{sw} = \frac{2\pi f \tau_{sw}(\varepsilon_{sw0} - \varepsilon_{sw\infty})}{1 + (2\pi f \tau_{sw})^2} + 0.475 \frac{\rho_b}{2\pi f \varepsilon_0} \frac{S_c^{m_v^{0.477}}}{m_v} \ (m_v \leq 0.3 \text{ m}^3/\text{m}^3 \text{ or } S_c \leq 20 \text{ g/kg}) \tag{15}$$

The L-band model has excellent performance when simulating the saline soil dielectric constant, and the model has an RMSE of 5.596 and an R$^2$ of 0.924 (as Figure 7). Compared with the C-band model, this model has a larger RMSE. Specifically, the RMSE of the model increased from 1.648 to 5.596. This result means that the accuracy of the model decreases for the L-band. The expressions and the simulated results of the two models were compared to obtain a deeper understanding of this model. The soil dielectric constant at the L-band has a relatively large value range, and this is the main reason leading to a higher RMSE than those obtained for the model-simulated result under the C-band; the RMSEs obtained under these two conditions are not comparable. However, the relative error is another quantitative index used to estimate the model-predicted results, and the L-band model

has the same degree of relative error as the C-band model. In addition, the error of the soil dielectric constant simulated by the aforementioned model has a similar distribution. Specifically, the dielectric constant has a larger error when the soil has high salinity and water content. In contrast, the model performs well when simulating the dielectric constant of soil with a low salinity and water content. This phenomenon may be due to the larger dielectric constant that occurs when soils contain high salinity and moisture. Generally, this study provides a model that can describe the relationship between soil properties and the dielectric constant at L-band.

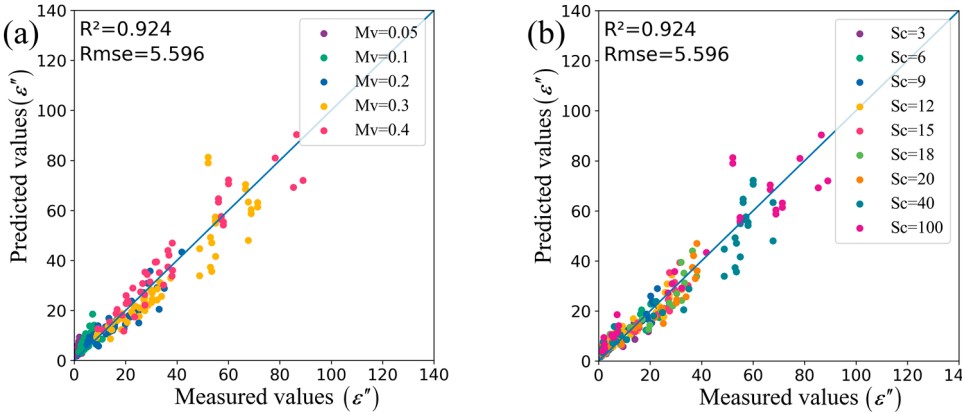

**Figure 7.** The L-band model-simulated result validation for different (**a**) soil moisture ($Mv$, $\mathrm{m}^3/\mathrm{m}^3$) and (**b**) salinity ($Sc$, g/kg) classifications.

The model represents the different formulas required for different microwave frequencies. Specifically, the empirical coefficients are related to the microwave frequency. Although the study gives specific model expressions for the C-band and L-band, these model expressions are inapplicable for other microwave frequencies. Therefore, future studies about saline soil dielectric constant should mention the empirical coefficient of the model. In further research, the change trends and patterns of the empirical coefficients should also be considered.

*3.3. Discussion*

Soils have different dielectric properties under different salinity and water content conditions [35], and the soil salinity mainly influences the imaginary part of the saline soil dielectric constant. The relationship between soil salinity and the conductivity of the soil solution needs to be represented using different expressions to simulate the soil dielectric properties under different salinity and water content degrees. Therefore, this study constructed a two-stage model to describe the imaginary part of the saltwater dielectric constant under different moisture and salinity ranges. However, the application conditions and the boundary of each model have not been fully confirmed.

The primary reason for the uncertainty of the model application range was the limitations of the soil samples. In this study, the saline soil dielectric constant of the soil sample dataset consisting of diverse soil moisture and salinity degrees was measured in the laboratory. Specifically, the soil samples had salinity contents of 0, 5, 10, 15, 20, 40, 100 g/kg, and water contents of 0.05%, 5%, 10%, 20%, 30%, and 40%. The salinity contents were not uniformly distributed between the low and high-value ranges. Specifically, there were many salinity degrees corresponding to soil samples with low salinity contents. In contrast, few salinity degrees were set under the high salinity ranges, especially when the soil salinity was greater than 20 g/kg. Although the soil salinity hardly exceeds 20 g/kg, the salinity degrees of soil samples have similar distributions to the real situation of the natural soil properties. However, these soil samples are insufficient for describing the soil dielectric properties with soil salinity changing in detail, especially for the soil situation with high salinity content. In this study, the improved model belongs to a semi-empirical

model, and the empirical coefficients were fitted using the experimental measurements of soil samples. The empirical coefficients are related to the soil samples, and the model has different empirical coefficients for different salinity ranges and water content ranges.

The soil samples may introduce uncertainty into the model, as the interval varies under different soil salinity contents. Therefore, soil samples with more salinity degrees need to be organized, and the soil salinity content interval should have a small value under high salinity conditions. Based on the soil sample datasets with refined soil properties derived through controlled and measured data, we will have the opportunity to deeply learn the soil dielectric properties and the mechanisms by which soil moisture and salinity act on the dielectric properties of soils.

## 4. Conclusions

The study found that the relationship between the soil dielectric constant and salinity is expressed differently for high- and low-salinity soils based on existing models and experimental measurements. Therefore, an improved model was established with a two-stage expression under different water contents and salinity degrees. Compared with the original model, the enhanced model performs better in simulating the soil dielectric constant, especially for soils with high moisture and salinity content. Specifically, the RMSE of the total soil samples decreases from 7.3 to 1.6, and the RMSE of the simulated dielectric constant significantly decreases when the soil salinity is greater than 10 g/kg. Moreover, the application conditions of the improved model are greatly expanded. The original model is applicable when the soil salinity is less than 20 g/kg, and the accuracy of the simulated results decreases quickly when the soil salinity and water content increase. Therefore, the enhanced model also performs well when the soil salinity is greater than 20 g/kg, and the accuracy is eventually applicable when the soil salinity is greater than 40 g/kg. Based on the C-band model, the study also gives the model expression for the L-band, and the L-band model also has a suitable accuracy. Generally, the model is significantly valuable for retrieving soil moisture and salinity datasets using microwave remote sensing data, especially for acquiring the above two essential surface parameters when they have large values. The model is valuable for acquiring spatial salinity and water content datasets at the global and regional scales, as is often required in research on wetlands, saline-alkali land, and salt lakes.

**Author Contributions:** Methodology: L.G. and X.S.; Validation: L.G. and X.S.; Formal analysis: L.G. and X.S.; Investigation: L.G. and X.S.; Writing—original draft preparation: L.G.; Writing—review and editing: L.G., X.S., J.M., X.L., P.L., W.W. and X.Z. All authors have read and agreed to the published version of the manuscript.

**Funding:** This research was funded in part by The Second Tibetan Plateau Scientific Expedition and Research (STEP) program "Changes in grassland ecosystem types, structure, productivity and driving factors in western Sichuan and northwest Yunnan" 2019QZKK0302-02, The Third Xinjiang Scientific Expedition Program (Grant No. 2022xjkk0402), The Second Tibetan Plateau Scientific Expedition and Research (STEP) program "Wetland Ecosystem and Hydrological Process Change" 2019QZKK0304-02.

**Data Availability Statement:** Data are contained within the article.

**Conflicts of Interest:** The authors declare no conflicts of interest.

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
