# Peer review of "An Enhanced Saline Soil Dielectric Constant Model Used for Remote Sensing Soil Moisture and Salinity Retrieval"

_remotesensing, doi:10.3390/rs16030452_

Round 1
Reviewer 1 Report
Comments and Suggestions for Authors
The authors didn't described the analysis methods for the soil's properties in 2.1 chapter. Also, the authors should describe the characteristics of the studied soils, both from the morpholofical, physical and chemical point of view. Also, it will be interested to be presented the location of the sampling. Moreover, the authors discuss sbout soil moisture, but it is not specified in the results chapter if the values presented are from the native soil conditions or it were obtained in a controlled laboratory conditions. Another aspect, in introduction chapter (line 45-47), authors discuss about about soil salinization as a threat to agriculture, but they do not specify anything about the actual context of climate change and are not any discussions about the climate conditions in the studied area.
Author Response
(1)The authors didn't described the analysis methods for the soil's properties in 2.1 chapter. Also, the authors should describe the characteristics of the studied soils, both from the morpholofical, physical and chemical point of view. Also, it will be interested to be presented the location of the sampling. Moreover, the authors discuss sbout soil moisture, but it is not specified in the results chapter if the values presented are from the native soil conditions or it were obtained in a controlled laboratory conditions.
Ans: Thanks for your comments. The soil moisture and salinity degrees about soil samples was controlled under the laboratory conditions. We have added the detailed explanation about soil samples collection and measurements in Section 2.1.
(2)In introduction chapter (line 45-47), authors discuss about soil salinization as a threat to agriculture, but they do not specify anything about the actual context of climate change and are not any discussions about the climate conditions in the studied area.
Ans: Thanks for your comments. This sentence aims to explain salinity is an essential soil parameter influencing land surface processes, such as the freezing and thawing cycle and cultivated land salinization. The soil salinity dataset is important to monitor soil salinization. The soil dielectric constant model is an important part to retrieve soil parameters using microwave. We should build the high-precision saline soil dielectric constant model to improve the performance of retrieval model. We have revised the related expression to clearly express the previous aims.

Reviewer 2 Report
Comments and Suggestions for Authors
Authors successfully improved the WYR especially saline soil relative permittivity model. RMSE of the simulated results decreased for soil with high salinity and water content. The improved model had higher coefficient of determination R2.
In part “2.1 Soil sample dataset”, measured soil samples, soil samples with different salinity degrees and different volumetric water contents were listed. But the total number of samples and the total number of measurements were not given.
Instead of referring to papers 37 and 35, it would be clearer to state whether the relative permittivity was measured or calculated from other values.
Manuscript contains many formal errors.
The quantities must be in italics (also Mv, Sc).
Line 65 – instead of “weighted”, right will be “mass”.
Lines 93, 97 – instead of e.g., "Wu, Wang, Zhao and Liu [37]”, better will be “Authors [37]” or “Wu et al. [37]”.
Lines 144, 145 and other – equations should be corrected, the horizontal fractional line should be used.
Line 146 and after all equations – according to Instructions for Authors – “where” should be flushed to the margin (without first line indentation).
Lines 161 and other – indexes must be repaired, e.g., instead of “
“, right will be “
”.
Lines 171, 202 and other – why some quantities are marked in a larger font?
Lines 204 and other equations – Equation 8 must be corrected, do not use the middle dot as a multiplication sign.
Line 205 – I recommend in the quantity of soil salinity content Sc, for c to be a subscript. When exponentiating the quantity Sc, it looks like only c is exponentiated.
Line 227 – instead of “determine coefficient (R2)”, better will be “coefficient of determination (R2)”.
Figures 1 and 3, equations12, 13, and other – the unit of Sc (g/kg) should also be in the figures when labelling the samples, and also at the limitations of the equations, not only in their descriptions.

Author Response
Reviewer2:
(1). In part “2.1 Soil sample dataset”, measured soil samples, soil samples with different salinity degrees and different volumetric water contents were listed. But the total number of samples and the total number of measurements were not given.
Ans: Thanks for your comments. We have added the detailed description about soil samples from Line 124 to 129.
(2). Instead of referring to papers 37 and 35, it would be clearer to state whether the relative permittivity was measured or calculated from other values.
Ans: Thanks for your comments. The complex dielectric constant of the saline soil samples was measured using a vector network analyzer. This is a widely-used instrument to measure soil dielectric constant. We have added the related explanation from Line 135 to 140.
(3). The quantities must be in italics (also Mv, Sc).
Ans: Thanks for your comments. We have carefully searched and adjusted quantities as italics in the all Equations.
(4). Line 65 – instead of “weighted”, right will be “mass”.
Ans: Thanks for your comments. We wish to clarify that "the dielectric constant of a mixed soil is equivalent to the weighted sum of the dielectric constants of its individual components." However, we did not encounter any references to "mass sum," so there are no adjustments made in response to this particular comment in our revisions.
(5). Lines 93, 97 – instead of e.g., "Wu, Wang, Zhao and Liu [37]”, better will be “Authors [37]” or “Wu et al. [37]”.
Ans: Thanks for your comments. We have adjusted the reference in Line 93 and Line 97.
(6). Lines 144, 145 and other – equations should be corrected, the horizontal fractional line should be used.
Ans: Thanks for your comments. We have carefully searched and corrected the equations using the horizontal fractional in Equation 1, 2, 8 and 10.
(7). Line 146 and after all equations – according to Instructions for Authors – “where” should be flushed to the margin (without first line indentation).
Ans: Thanks for your comments. We have carefully searched and delete the first line indentation before the “where” in Line 147, 163, 171, 180, 205, 210, 213.
(8). Lines 161 and other – indexes must be repaired, e.g., instead of “ “, right will be “”.
Ans: Thanks for your comments. We have carefully searched and repaired some indexes in the revised version.
(9). Lines 171, 202 and other – why some quantities are marked in a larger font?
Ans: Thanks for your comments. We have adjusted quantities as the same size in the revised version.
(10). Lines 204 and other equations – Equation 8 must be corrected, do not use the middle dot as a multiplication sign.
Ans: Thanks for your comments. We have carefully searched and removed the middle dot in Equation 8, 9, 10 , 11, 12 and 13.
(11). Line 205 – I recommend in the quantity of soil salinity content Sc, for c to be a subscript. When exponentiating the quantity Sc, it looks like only c is exponentiated.
Ans: Thanks for your comments. We have adjusted “c” to be a subscript in the revised version.
(12). Line 227 – instead of “determine coefficient (R2)”, better will be “coefficient of determination (R2)”.
Ans: Thanks for your comments. We have replaced “determine coefficient” with “coefficient of determination” in Line 227.
(13). Figures 1 and 3, equations12, 13, and other – the unit of Sc (g/kg) should also be in the figures when labelling the samples, and also at the limitations of the equations, not only in their descriptions.
Ans: Thanks for your comments. we have added the unit of Sc (g/kg) at the limitations of the equations. We tried adding units to the Figures, but the end performance wasn't good, so we didn't add this unit in the Figures 1 and 3.

Reviewer 3 Report
Comments and Suggestions for Authors
No comments to the current version
Comments on the Quality of English LanguageNo comments to the current version
Author Response
Reviewer3:
Ans: Thanks for your comments. We have carefully searched and revised multiple minor issues in the this version.

Reviewer 4 Report
Comments and Suggestions for Authors
This paper found that the relationship between the soil dielectric constant and salinity
is expressed differently for high- and low-salinity soils based on existing models and experimental measurements. Therefore, an improved model was established with a two stage expression under different water contents and salinity degrees. The structure of the manuscript is a little confusing, and there are some questions that need to be clarified and improved.
Major comment
1. First of all, I think the innovation point of this paper is very small, only the parameter optimization for different soil moisture and salinity of the original model. The author needs to elaborate on his innovations.
2. The author established with a two stage expression under different water contents and salinity degrees. This model is only based on certain measured data, and is a small amount of data published by others, its innovation and the scope of application of the model are still questionable.The author should explain why the data of these two articles were chosen, and the applicability of the model in other regions.
3. In terms of structure and narration, this paper is not ready for publication.
4. First of all, the introduction needs to be rewritten. The introduction of this article is not easy to grasp the point, and it always rambles a little bit here and there. It needs to be more clearly organized
5. The second part of the paper also needs to be rewritten, which should first introduce the method, then introduce the study area and data collection, and finally introduce the model evaluation method.
6. In the third part of Results, the simulation results of several models can be compared.
7. Line 299,“If the soil moisture is more than 0.3 m3/m3 and salinity is more than 20 g/kg”,Here is 0.2 or 0.3?
Comments on the Quality of English LanguageThis paper found that the relationship between the soil dielectric constant and salinity
is expressed differently for high- and low-salinity soils based on existing models and experimental measurements. Therefore, an improved model was established with a two stage expression under different water contents and salinity degrees. The structure of the manuscript is a little confusing, and there are some questions that need to be clarified and improved.
Major comment
1. First of all, I think the innovation point of this paper is very small, only the parameter optimization for different soil moisture and salinity of the original model. The author needs to elaborate on his innovations.
2. The author established with a two stage expression under different water contents and salinity degrees. This model is only based on certain measured data, and is a small amount of data published by others, its innovation and the scope of application of the model are still questionable.The author should explain why the data of these two articles were chosen, and the applicability of the model in other regions.
3. In terms of structure and narration, this paper is not ready for publication.
4. First of all, the introduction needs to be rewritten. The introduction of this article is not easy to grasp the point, and it always rambles a little bit here and there. It needs to be more clearly organized
5. The second part of the paper also needs to be rewritten, which should first introduce the method, then introduce the study area and data collection, and finally introduce the model evaluation method.
6. In the third part of Results, the simulation results of several models can be compared.
7. Line 299,“If the soil moisture is more than 0.3 m3/m3 and salinity is more than 20 g/kg”,Here is 0.2 or 0.3?
Author Response
Reviewer4:
(1) The introduction needs to be rewritten. The introduction of this article is not easy to grasp the point, and it always rambles a little bit here and there. It needs to be more clearly organized.
Ans: Thanks for your comments. We have rewritten the introduction part. Specifically, this part firstly introduce the importance of soil moisture and soil salinity, the second paragraph introduce the microwave remote sensing is the important pathway to acquire spatial soil moisture and salinity, and soil dielectric constant model is the import part in the microwave soil moisture retrieval model. Fianlly, we review the each specific saline soil dielectric constant model and the characteristics of each model. Based on the advantages and disadvantages of these model, we find the problem of existing models and corresponding objective of this study. We have clearly organized this section in the revised version.
In addition, based on the soil samples measurements and validation about the existing model, the study found that the soil dielectrci reflected different characteristic under diverse salinity and water contet. This is an important found about the nature of soil dielectric feature. The manuscript built a two-stage model based on this Foundation. We have further elaborated innovation in the section about Conclusion and the objective of the Manuscript.
(2)The second part of the paper also needs to be rewritten, which should first introduce the method, then introduce the study area and data collection, and finally introduce the model evaluation method.
Ans: Thanks for your comments. We have rewritten the second part of the paper to clearly express the soil samples. However, the manuscript is about the model builting, the characteristics of the study area have limited impact on the model's performance. The manuscript aim is to ensure the model's applicability across diverse scenarios. The water content and salinity degrees distributions of soil samples is important for the finally model performance, and we have detailed the relatered part in the revised version.
(3). In the third part of Results, the simulation results of several models can be compared.
Ans: Thanks for your comments. We have already done a series of work on the saline soil dielectric properties, of which the modeling of the dielectric constant here is only a part. We have compared the simulation results of these several models in the previous works. The detailed see “Liang Gao, Xiaoning Song, Pei Leng, Jianwei Ma,Xinming Zhu, Ronghai Hu, Yanfen Wang, Yanan Zhang, Dewei Yin. Impact of Soil Salinity on Soil Dielectric Constant and Soil Moisture Retrieval From Active Microwave Remote Sensing. IEEE Transactions on Geoscience and Remote Sensing, 2022, 60: 1-12”. In this study, we found the WYR models has the best performance, so we do some comparation with this model simulation. We have added the explanation from Line 196 to Line 202.
(4). Line 299,“If the soil moisture is more than 0.3 m3/m3 and salinity is more than 20 g/kg”,Here is 0.2 or 0.3?
Ans: Thanks for your comments. This limitations of the equations are 0.3 m3/m3. The manuscript reveals that the original model's validation results exhibit subpar performance when soil moisture exceeds 0.2. Upon further analysis, it became evident that the original model was calibrated using empirical coefficients derived from all measurements. Soil dielectric properties exhibit varying characteristics across different moisture and salinity ranges, which significantly influences the model's performance. Ultimately, we validated the model's applicability based on changes in soil dielectric constants, setting the threshold at 0.3. Concurrently, the model demonstrates optimal performance when adhering to this equation limitation.

Round 2
Reviewer 1 Report
Comments and Suggestions for Authors
I thank the authors for considering the suggestions regarding the scientifical paper.
Reviewer 4 Report
Comments and Suggestions for Authors
Accept in present form